# Obesity-Linked PPARγ Ser273 Phosphorylation Promotes Beneficial Effects on the Liver, despite Reduced Insulin Sensitivity in Mice

**DOI:** 10.3390/biom13040632

**Published:** 2023-03-31

**Authors:** Maiara Ferreira Terra, Marta García-Arévalo, Thayná Mendonça Avelino, Karina Y. Degaki, Murilo de Carvalho, Felipe Rafael Torres, Angela Saito, Ana Carolina Migliorini Figueira

**Affiliations:** 1Brazilian Biosciences National Laboratory (LNBio), Brazilian Center for Research in Energy and Materials (CNPEM), Campinas 13083-100, Brazil; 2Post Graduate Program in Functional and Molecular Biology, Institute of Biology, State University of Campinas (Unicamp), Campinas 13083-872, Brazil; 3Post Graduate Program in Pharmacological Science, State University of Campinas (Unicamp), Campinas 13083-872, Brazil; 4Brazilian Synchrotron Light Laboratory (LNLS), Brazilian Center for Research in Energy and Materials (CNPEM), Campinas 13083-100, Brazil

**Keywords:** insulin sensitization, PPAR, PPAR phosphorylation, type 2 diabetes

## Abstract

Since the removal of thiazolidinediones (TZDs) from the market, researchers have been exploring alternative anti-diabetic drugs that target PPARγ without causing adverse effects while promoting insulin sensitization by blocking serine 273 phosphorylation (Ser273 or S273). Nonetheless, the underlying mechanisms of the relationship between insulin resistance and S273 phosphorylation are still largely unknown, except for the involvement of growth differentiation factor (GDF3) regulation in the process. To further investigate potential pathways, we generated a whole organism *knockin* mouse line with a single S273A mutation (KI) that blocks the occurrence of its phosphorylation. Our observations of KI mice on different diets and feeding schedules revealed that they were hyperglycemic, hypoinsulinemic, presented more body fat at weaning, and presented an altered plasma and hepatic lipid profile, distinctive liver morphology and gene expression. These results suggest that total blockage of S273 phosphorylation may have unforeseen effects that, in addition to promoting insulin sensitivity, could lead to metabolic disturbances, particularly in the liver. Therefore, our findings demonstrate both the beneficial and detrimental effects of PPAR S273 phosphorylation and suggest selective modulation of this post translational modification is a viable strategy to treat type 2 diabetes.

## 1. Introduction

Type 2 diabetes (T2D) is an increasingly prevalent global health challenge in the 21st century [1]. Obesity, a major risk factor for developing T2D, has become a worldwide epidemic due to rapid urbanization, lifestyle changes, dietary habits, and sedentary behavior [2]. T2D is characterized by insulin resistance and β-cell failure and has been treated with thiazolidinediones (TZDs) for many years due to their ability to induce insulin sensitivity and reduce glycemia [3,4].

The underlying mechanism of TZDs remained a mystery until the mid-1990s when the ligands for peroxisome proliferator-activated receptor gamma (PPARγ) were identified [5]. PPARγ, a member of the nuclear receptors’ superfamily, which is primarily expressed in white adipose tissue (WAT), is a transcription factor and is considered the master regulator of adipocyte differentiation [6]. It is also the main receptor for TZDs, and certain compounds can bind to it with high affinity, becoming fully activated in terms of transcriptional response due to the strong agonism caused by some of the TZDs.

However, the reported adverse effects of TZDs, particularly rosiglitazone, such as fluid retention, edema, heart failure, and weight gain [7,8,9], motivated a thorough investigation of the effects of PPARγ activation by full agonists [4]. As a result, many TZDs were removed from the market or used under restriction.

Subsequently, phosphorylation of PPARγ at serine 273 (S273) was identified and linked to increased insulin resistance. It was suggested that the insulin sensitization induced by many anti-diabetic drugs was due to the blockade of this phosphorylation, caused by conformational modifications induced by ligand binding [10]. Consequently, researchers have focused on molecules that can promote this blockade without activating the nuclear receptor, resulting in improved insulin sensitivity without the development of adverse effects [11,12,13,14]. Additionally, some ligand binding protection mechanisms against phosphorylation have been proposed [11,15,16].

The blockage of S273 by mutating it to alanine (S273A) in in vitro assays has been reported to promote increased activation of PPARγ, as well as the recruitment of coactivators and the dissociation of corepressors [17]. Moreover, a mouse lineage with PPARγ S273A mutation was generated via homologous recombination in mouse embryonic stem cells (mESCs), in 2020, and has demonstrated protection against the development of insulin resistance in response to a high-fat diet (HFD). The improvement of insulin sensitivity was attributed to the significant reduction in growth differentiation factor 3 (Gdf3) levels in these animals, which were also shielded from the adverse effects of TZDs [18].

In this context, to investigate the functional and molecular mechanisms involved in the blockage of S273 phosphorylation, we employed the CRISPR/Cas9 technique to develop a *knockin* lineage with an S273A mutation. Our focus was on lipid and hepatic metabolisms, as well as liver gene expression, with the aim of elucidating the effects of this single mutation on the whole organism from weaning and after 8 and 18 weeks of feeding with different diets.

## 2. Material and Methods

### 2.1. Generation of PPARy S273A Knockin Mice and Genotyping

The PPARγ S273A (c.817T > G p.S273A) mouse line was generated through CRISPR/Cas9 genome editing tool by Model Organism Laboratory at LNBio/CNPEM. The single guide RNA (sgRNA) was generated using a 20 nucleotide (nt) guide sequence, specific for the fifth exon of the *Nr1c3* gene, as proposed by [19] and [20], and cloned into px330 vector in fusion with the transactivating crRNA (tracrRNA). Molecules of sgRNA and *S. pyogenes* Cas9 mRNA were in vitro transcribed using MEGAshortscript T7 Transcription Kit (Invitrogen) and mMESSAGE mMACHINE T7 Transcription Kit (Invitrogen), respectively, and purified using MEGAclear Transcription Clean-Up kit (Invitrogen). A single-stranded oligodeoxynucleotide (ssODN) donor was designed to contain S273A missense mutation, one point mutation to create a ScaI restriction enzyme site in the intron 5, and three silent mutations to avoid cleaving DNA after homologous recombination. ssODN (200 bases) was purchased from IDT DNA, designed to be complementary to the non-target strand and asymmetrical, in which 162 homology bases were distributed as following 1/3 on the PAM-distal side and 2/3 on the PAM-proximal side [21] (Appendix A).

The sgRNA (50 ng/µL), Cas9 mRNA (50 ng/µL), and ssODN (100 ng/µL) were microinjected into the pronuclei of C57BL/6J zygotes. Microinjected embryos were implanted into pseudopregnant CB6F1 (BALB/c × C57BL/6J) foster mothers. Mice were genotyped by PCR amplification from tail genomic DNA using PPARγ-PCR forward primer (5′-TCAGGTTTGGGCGGATGCCA-3′) and PPARγ-PCR reverse primer (5′-TGTGCCCACTTTGGACCTGGG-3′), followed by screening of animals by T7 Endonuclease I assay (T7EI) and/or ScaI restriction enzyme digestion.

For T7EI assay, 627 bp PCR amplicon was purified by PCR purification kit (Qiagen, Singapore) and submitted to the assay with the T7 Endonuclease I enzyme (New England Biolabs, Ipswich, MA, USA), which recognizes and cleaves non-perfectly matched DNA. Briefly, 300 ng of founder’s PCR fragment was denatured at 95 °C for 10 min, followed by cooling annealing at –2 °C/second until 85 °C, and slow cooling at −0.1 °C/second until 25 °C. After the formation of DNA heteroduplexes, they were incubated with 5 U of T7EI enzyme (New England Biolabs) at 37 °C for 30 min. Then, the reaction was resolved in a 1% agarose gel electrophoresis [22]. Purified PCR amplicons digested by ScaI enzyme were also submitted to Sanger sequencing to confirm the insertion of desired mutations. Their undigested fragment must contain 627 bp and, after cleavage, bands of approximately 427 bp and 200 bp should be observed (Appendix A).

The *knockin* (KI) animals were mated with wild-type C57BL/6J, and heterozygous littermates were intercrossed to generate both *knockin* PPARγ S273A and wild-type (WT) mice, which originated the next offspring; and animals since the third generation of homozygous were used in the experiments. All animal procedures were approved by the Institutional Animal Care and Use Committee (CEUA/CNPEM, protocol numbers 72 and 80).

### 2.2. Animals

Three and six-week-old male KI and WT mice, weighing between 10 and 18 g, were housed in the pathogen-free animal facility at the Model Organism Laboratory, at Brazilian Biosciences National Laboratory (LNBio, Campinas, Brazil) inside the Brazilian Center for Research in Energy and Materials (CNPEM, Campinas, Brazil). All animals were maintained at a number of 3 to 5 animals per cage, with a photoperiod of 12:12 light/dark cycle, at 21–24 °C, with free access to food and water. All procedures were conducted in accordance with the recommendations of the National Institutes of Health’s Guide for the Health and Use of Laboratory Animals and were approved by the Ethics Animal Use Committee of CNPEM (CEUA/CNPEM), with protocol numbers 72 and 80. All efforts were made to minimize suffering.

### 2.3. Diets and Experimental Design

Animals were divided into two groups based on their age: one started diet feeding immediately after weaning (3 weeks old) and the other, at 6 weeks old. For the first one, male mice were fed since weaning with a control (MNT-AIN93M PragSoluções, Jaú, SP-Brazil) or a high-fat high-sucrose diet (HFHS-42% as fat, PragSoluções, Jaú, SP-Brazil), for 8 weeks, until eleven weeks old (diets’ compositions in Appendix A). For the second group, six-week-old male mice were fed with high-fat diet (HFD-60% as fat, PragSoluções, Jaú, SP-Brazil-Appendix A) for 18 weeks, until 24 weeks-old. Animals and diets were weighed every week and we performed measurements of weight, glucose and plasma insulin levels of animals at weaning, and in the two groups, for a number of 6–8 animals.

On the euthanasia day, which varied from fed state until 12 h fasted, blood samples were collected by decapitation using heparin, with the collection of trunk blood, which was centrifuged (1200× *g* for 20 min at 4 °C). Plasma was stored at −80 °C for measurement of insulin, cholesterol, and triglycerides. Epidydimal white adipose tissue (eWAT), brown adipose tissue (BAT), liver, heart, kidney, gastrocnemius muscle, and tibia were dissected and weighed. Samples of liver and eWAT were processed for histological analysis, and the other organs were stored at −80 °C for further analysis. Liver samples were also used for determination of hepatic cholesterol, triglycerides, and glycogen content (details of experimental design are presented in Appendix A).

### 2.4. Insulin Tolerance Test (ITT) and Glucose Tolerance Test (GTT)

At the end of 8 weeks of MNT and HFHS feeding, a long insulin tolerance test (ITT) was performed, using 0.75 UI/kg of insulin, and blood samples were collected from the dorsal tail vein. The glucose measurements were performed with the Accu-chek Performa (Roche, Basel, Switzerland) glucometer, at the times: 0, 15, 30, 45, and 60 min. In addition, after 18 weeks of HFD feeding, short ITT was performed by measuring glycemia at 0, 4, 8, 12, 16, and 24 min. Glucose disappearance rate (KITT) was measured as described previously [23]. Glucose tolerance test (GTT) was also performed at the end of 8 weeks of MNT and HFHS diets, after 12 h of fasting, applying an intraperitoneal glucose dose of 2 g/kg and measuring glycemia at 0, 15, 30, 60, and 120 min.

### 2.5. Plasma Biochemical Analysis

Tail blood samples in fed state were collected with heparin, and plasma was collected after blood centrifugation at 2000× *g* for 20 min, at 4 °C. Insulin was measured by ELISA kit (Ultrasensitive Mouse Insulin #90080-Crystal Chem, Elk Groove Village, IL, USA), and cholesterol and triglycerides were analyzed using enzymatic assay kits (LaborLab Liquid Stable-# 1770290-LaborLab, Guarulhos, SP, Brazil), according to the manufacturers’ instructions. Adiponectin was measured by ELISA kit (Adiponectin Mouse #80569—Crystal Chem, Elk Groove Village, IL, USA).

### 2.6. Liver Histology

Hepatic caudate lobe was removed, dissected, fixed in 4% paraformaldehyde for 24 h, dehydrated, embedded into a paraffin block, cut into 5 µm sections, and stained with hematoxylin-eosin by standard procedures for histological analysis. The images of each tissue slice were captured with a digital camera on a light microscope (Leica FS DM6, Wetzlar, Germany), with a magnification of 20Xx Five distinct regions were captured of each liver slide, with three slices, separated by 50 µm, for each animal. Only HFD-fed animals were analyzed by the segmentation method using Weka Segmentation and quantification with MorphoLibJ Plugin from Fiji.

### 2.7. eWAT Histology

Right eWAT were dissected and fixed in 4% paraformaldehyde for 24 h. Afterwards, they were dehydrated, embedded into a paraffin block, cut into 5 µm sections, and stained with hematoxylin-eosin by standard procedures for histological analysis. The brightfield images of each tissue slice were captured with a digital camera mounted on an upright light microscope (Leica FS DM6), using 10x objective. The whole tissue was captured in a panoramic picture to analyze all the adipocytes. To perform shape analysis (i.e., area and perimeter) for every single cell on each treatment (*n*), we developed a Macro workflow on Fiji for image segmentation and adipocyte identification, which were posteriorly used as input for a Python optimized script, as described in [14].

### 2.8. Hepatic Triglyceride Content Quantification

Furthermore, 100 mg of frozen liver was homogenized in a Polytron (PT1200 E), in 4:3:0.8 mixture of chloroform:methanol: PBS, and then centrifuged to obtain the organic layer [12]. The triglyceride content was measured from this layer using the Triglycerides LaborLab GODPAP Liquid Stable Kit (LaborLab—SP, Brasil), at 505 nm wavelength, in the multi-mode plate reader (Enspire-PerkinElmer, Singapore) following the manufacturer’s instructions [14].

### 2.9. Hepatic Glycogen Measurement

Hepatic glycogen content was measured using 300–500 mg of tissue, which was hydrolyzed by using 30% of potassium hydroxide (KOH) in water, at 95–98 °C, for 20 min. Afterwards, the solution was mixed by vortex, maintained on ice, and 10 µL was used to quantify total protein content through Pierce 660 nm Protein Assay Reagent (Thermo Scientific, Waltham, MA, USA) in a multi-mode plate reader (Enspire-PerkinElmer) (Adapted from [24,25]).

### 2.10. RNA Isolation, cDNA Synthesis, and Quantitative Real-Time PCR Analysis

TRIzol reagent (Ambion-Thermo Fisher Scientific,Waltham, MA, USA) and Polytron (PT1200 E) were used to extract the total ribonucleic acid (RNA) from liver, eWAT, and BAT, followed by the chloroform-isopropanol extraction method. RNA concentration and quality were checked by Nanodrop 2000 (ThermoScientific, 260/280 ratio higher than 1.8) and cDNA synthesis was carried out using 1 µg of RNA with the High-Capacity cDNA Reverse Transcription kit (Applied Biosystems, Waltham, MA, USA). Gene expression was evaluated by quantitative real time PCR (qPCR) reactions using SYBR Real Time PCR master mixes (ThermoFischer Scientific) in a 7500 Real Time PCR system (Applied Biosystems). The ΔΔ-Ct evaluation method [26] was used for data normalization, with expression of reference genes *36b4* and *Rpl27* [14]. The primer sequences used in this study are presented in Appendix A.

### 2.11. Western Blotting

Proteins from eWAT were extracted by macerating the tissue with a Polytron (PT1200 E) in a solution composed of 0.5 M EDTA 0.02% (*v*/*v*) pH8, 0.1% (*v*/*v*) 1 M Tris pH7.5, 0.0045% (*m*/*v*) sodium fluoride, 0.0019% (*m/v*) sodium orthovanadate, 10% Triton X-100, 0.02% (*v*/*v*) PMSF (serine protease inhibitor), 0.04% (*v*/*v*) Cip 25x (protein inhibitor cocktail), and water q.s.p., incubating it for 1 h on ice. Then, samples were centrifuged, and supernatant was collected for protein quantification using Pierce 660 nm Protein Assay Reagent (Thermo Scientific ). Electrophoresis gel was performed with 15 µg of protein and the following primary and secondary antibodies were used: anti-PPARγ (dilution 1:1000, Anti-PPARγ Antibody (E-8): sc-7273-Santa Cruz Biotechnology, Inc., Dallas, TX, USA), anti-S273 PPARγ (dilution 1:200, BS-4888R, Bioss, Boston, MA, USA), anti-vinculin (dilution 1:1000, ab18058-Abcam, Waltham, MA, USA), anti-rabbit IgG (dilution 1:5000, A0545, Sigma-Aldrich, St. Louis, MO, USA), and anti-mouse IgG (dilution 1:5000, 401253, Calbiochem, San Diego, CA, USA). The final result was obtained using peroxidase reaction through Amersham ECL Prime Western Blotting Detection Reagent-GE in the ImageQuant LAS 500 (GE Health Care Life Sciences, Chicago, IL, USA) equipment, as described [14].

### 2.12. Statistical Analysis

To perform statistical analysis, we used GraphPad Prism 8.0 statistical package and all results are presented as mean ± SEM. Analyses were performed using non-parametric *t*-test or multiple comparisons one-way ANOVA followed by Dunnett’s post hoc test for comparing the means of two or multiple groups, respectively. ITT test was analyzed by a two-way ANOVA followed by Dunnett’s post hoc test. Statistical significance was noted when *p* < 0.05.

Statistical analysis of WAT was performed on R-Studio, considering as outliers the values higher than the IQR (interquartile range) multiplied by 1.5, which were excluded from analysis. Descriptive statistics and statistical tests ‘ggstatsplot’ library considered non-parametric distribution (Wilcoxon rank sum test). This test considers H0 (null) and H1 (alternative) hypothesis. Bayesian tests were also implemented due to large sample number, where the Bayes factor (BF_10_) gives the evidence for H1 over H0. Higher BF_10_ indicates higher evidence in favor of alternative hypothesis, which means slighter and stronger difference between groups (between 3 and 20 is positive, 20 and 100 is strong, and over 100 is extremely strong) [14,27,28].

## 3. Results

### 3.1. Knockin Lineage Had Higher Body Weight, Hyperglycemia, and Low Insulin Levels at Weaning

To study the physiological effects of S273 phosphorylation blockage in the context of insulin sensitivity, we developed an S273A *knockin* (KI) lineage using CRISPR/Cas9 technique, which had a single nucleotide mutation (Ser > Ala) on PPARγ (Appendix A). Animals were genotyped at weaning by sequencing PPARg gene at the region of S273, and by restriction enzyme assay to confirm the presence or absence of mutations (Appendix A). At weaning, pups from litters of 6–8 animals of WT and KI groups were weighed, the glucose was measured using a glucometer, and plasma insulin levels were quantified by ELISA assay. Results show increased body weight, hyperglycemia, and reduced plasma insulin levels on KI lineage, indicating that S273A mutation modifies the metabolism of young animals, causing hypoinsulinemia (Figure 1A–C).

### 3.2. 8 Weeks of MNT Diet Increased Food Intake, Altered Plasma Metabolites, and Promoted Liver Modification

Maintenance diet (MNT) was administered to KI animals post-weaning for a period of eight weeks to ensure normal health and provide basic nutrients. During this period, although body weight gain remained unchanged, food intake increased (Figure 2A and Appendix A) in KI animals. No changes in organ weight were observed upon euthanasia (Figure 2B).

As expected, lean wild-type animals did not present phosphorylation at PPARγ S273, which is induced by obesity [10], and no alterations in intraperitoneal insulin (ipITT) and glucose (ipGTT) tolerance tests were detected (Appendix A). However, KI animals showed increased glycemia in both fasted and fed states (Figure 2C,D), presenting increased plasma insulin levels in the fed state (Figure 2E) and no changes in this hormone in the fasted state (Figure 2F). In terms of plasma lipid levels, only an increase in triglycerides in KI-fed animals was observed (Figure 2G and Appendix A). Histological analysis of livers from KI animals revealed an altered hepatocyte morphology, with the presence of fat and an apparent increased cellular volume (Figure 2I). Gene expression analysis showed increased expression of *TNFα*, *Cd36*, and *Fabp4*, while *PPARγ* decreased, indicating inflammation, liver damage, and lipotoxicity (Figure 2H). Moreover, decreased hepatic glycogen content was observed in both fasted and fed states of KI animals (Figure 2J,K), as well as no changes in liver lipids content (cholesterol and triglycerides—Appendix A). Collectively, these results demonstrate that even with a regular diet, PPARγ S273A mice are hyperglycemic, hyperinsulinemic, and present high levels of triglycerides; however, they remain non-insulin resistant and/or glucose intolerant, despite displaying signs of liver injury.

### 3.3. 8 Weeks of HFHS Increased WAT, Promotes Liver Damage, but Was Not Sufficient to Promote Insulin Resistance and Glucose Intolerance

To evaluate the effects of an obese condition a high-fat high-sucrose (HFHS) diet was administered on wild-type (WT) and *knockin* (KI) animals for eight weeks. Results showed that the WT and KI animals did not differ significantly in terms of food intake or body weight gain, not developing obesity (Appendix A). At euthanasia, the KI animals displayed increased white adipose tissue (WAT) weight and decreased liver and brown adipose tissue (BAT) weights, with no alterations in gastrocnemius muscle (Figure 3A).

Additionally, the 8-week HFHS diet did not lead to the development of glucose intolerance or insulin resistance in either group (Figure 3B,C), nor were changes observed in glycemia or plasma insulin levels in both fasted and fed states (Appendix A). Moreover, the KI animals had decreased plasma triglycerides only in the fed state (Figure 3D and Appendix A). In liver histology, more lipid droplet deposition was observed in the KI animals (Figure 3E), as well as decreased expression of peroxisome proliferator-activated receptor gamma (*PPARγ*) and fatty acid binding protein 4 (*Fabp4*) levels (Figure 3H). Further, KI animals exhibited increased hepatic glycogen content in the fasted state and decreased in the fed state (Figure 3F,G, respectively), with no changes in the hepatic lipid profile (Appendix A).

### 3.4. 16 Weeks of HFD Promoted Less Weight Gain, Insulin Sensitivity, Changes in PPARγ Expression, Liver Injury, and Adipocytes Hypertrophy in KI Animals

The effects of a prolonged high-fat diet (HFD) on WT and KI animals were observed when the animals were 6 weeks old and fed with HFD for 18 weeks. Results showed that the KI lineage had lower body weight gain despite consuming the same amount of HFD as the WT animals (Appendix A). By the end of 16 weeks of HFD feeding (week 22), both groups had achieved the same weight, with WT animals displaying significantly higher body weight from week 15 onwards, indicating reduced weight gain in KI animals in the last weeks of feeding, and faster weight gain in the WT group (Appendix A). In adulthood (24 weeks old), the KI animals exhibited improved insulin sensitivity with increased glucose disappearance rate (KITT) (Figure 4A,B), reduced plasma insulin levels, and increased adiponectin (Figure 4C,D, respectively), in addition to unchanged glycemia levels (Appendix A).

In epididymal white adipose tissue (eWAT), decreased *PPARγ* gene expression was observed for both WT and KI animals due to HFD feeding (Figure 4E). Interestingly, after a regular diet, the KI animals still presented reduced *PPARγ* gene expression compared to the WT, suggesting that, under the experimental conditions, phosphorylation prevention may reduce this nuclear receptor expression, even under a regular diet. The decreased PPARγ content caused by HFD was observed in both protein content and mRNA gene expression in the WT group. In contrast, the KI animals showed a unique behavior after HFD, displaying higher levels of PPARγ S273A protein content (Figure 4F,G and Appendix A), in contrast to lower gene expression.

Considering S273 phosphorylation levels, our findings revealed that HFD induces an increase in PPARγ phosphorylation in WT mice, but not in KI mice, as expected (Figure 4H). However, the complete absence of S273 phosphorylation in KI was not observed and might be attributed to the partial specificity of the used antibody for this serine residue of PPARγ under our experimental conditions.

Moreover, after HFD, the phosphorylation blockage in KI led to increased liver steatosis (Figure 4I,J), reduction of hepatic triglycerides content (Figure 4K), and of liver weight (Appendix A). All these results also point to liver damage in KI animals fed with HFD, providing further evidence that total phosphorylation prevention may be detrimental to this organ.

When examining WAT histological images of mice fed with HFD, we observed that KI animals presented increased adipocytes area and perimeter (Figure 5A). The image analysis showed that *knockin* animals increased the adipocytes’ sizes to intermediate and large ones (2500 < area < 7500 µm^2^ and perimeter > 500 µm (Figure 5B,C and Appendix A)), while WT mice had four times more small and intermediary cells in comparison to KI. In this sense, we can consider that KI animals presented adipocyte hypertrophy, while the WT group presented adipocyte hyperplasia after HFD.

The gene expression in eWAT also showed differences regarding the inhibition of S273 PPARg phosphorylation. The *Nr1d2* was decreased in KI animals, as well as *PPM1A*, while there was an increase in *Gdf3* expression (Figure 5D). Regarding the expression of genes related to adipogenesis, lipid and glucose metabolism (Figure 5E), we observed that KI animals had decreased expression of *Pdk4*, *Fabp4*, and *Cd36*. However, analysis of *Ucp1*, a thermogenic marker, revealed that this group had increased expression of this gene in both WAT and BAT (Figure 5F).

## 4. Discussion

The relation between PPARγ and insulin resistance has been extensively studied, particularly since the discovery of its S273 phosphorylation. Despite the numerous investigations into the mechanisms of action of this specific receptor’s phosphorylation, they remain unclear, and researchers are focused on finding answers for two principal properties of PPAR: the impediment of S273 phosphorylation to induce insulin sensitivity, and the promotion of selective agonism to avoid adverse effects. In this scenario, our study aimed to evaluate the physiological effects of a whole organism with PPARγ S273 phosphorylation blockage by mutating it to alanine through the CRISPR/Cas9 system.

In our initial analysis, we evaluated the animals at weaning and observed that the KI animals presented higher body weight and hyperglycemia, as well as low levels of insulin. While we expected to see higher levels of insulin due to the increased glycemia observed [29], the measured low rates of this hormone may be related to several factors, such as decreased β-cell glucose metabolism, islet steatosis, reduced β-cell mass and proliferation, increased glycogen rates, or depletion of insulin stores [30,31]. Our data indicate that blocking S273 phosphorylation in animals at this age results in compensation for hyperglycemia without causing hyperinsulinemia, suggesting an underlying hormonal imbalance. Furthermore, it indicates that carbohydrates were not properly processed, leading to glucose accumulation without provoking insulin resistance. Regarding food intake, we observed that KI animals ingested more food than the WT animals, suggesting dysregulation of leptin, neuropeptide Y, and adiponectin pathways, which is directly related to PPARγ activation and have been found to be modified by this receptor’s phosphorylation [32].

In a second evaluation, we investigated the consequences of the S273A mutation in WT and KI animals at an advanced age, under normal and high-fat high-sucrose (HFHS) diet conditions. The objective was to understand the effects of the mutation on regular healthy nutrient and obese conditions. Animals with the MNT diet exhibited increased food intake but no changes in body weight gain or organ weight. These changes in feeding behavior and weight maintenance were related to the leptin pathway [33], suggesting that phosphorylation could interfere with it, as previously suggested [32].

We found no changes in ipITT and ipGTT regarding glucose metabolism, as expected since both WT and KI animals are lean and phosphorylation of S273 is induced by obesity [10]. Therefore, neither group developed insulin resistance or glucose intolerance. However, KI animals developed hyperglycemia in both fed and fasted states after MNT diet. This behavior may be explained by increased hepatic glycogenolysis, as observed through reduced hepatic glycogen in both fed and fasted states for KI animals, promoting glucose release, increasing its plasma levels, and consequently, increasing plasma insulin levels to control hyperglycemia [34,35].

Additionally, liver histology showed morphologically distinct hepatocytes in KI animals, suggesting liver damage. The changes in hepatic gene expression, with reduced levels of *PPARγ* and increased levels of *TNFα*, *Fabp4*, and *Cd36*, supported this assumption.

Previous studies have suggested that a diet consisting of a specific proportion of macronutrients, sulfur-containing amino acids, and lipotropic agents (known as the MNT diet, based on the AIN-93M formula) can lead to increased hepatocyte volume and fatty liver, as well as changes in serum markers such as cholesterol, glucose, and triglycerides [36,37,38]. Our study aimed to investigate the effects of the MNT diet on KI animals and found that it had a more pronounced impact on this group. Specifically, increased levels of TNFα in the KI animals indicated the presence of an inflammatory environment and changes in insulin pathway [39,40]. The observed increased levels of *Fabp4* is related to the development of non-alcoholic fatty liver disease (NAFLD) [41,42,43], which involves *Cd36* expression as a potential marker, and hepatic lipotoxicity [44,45]. Finally, the increased plasma triglycerides levels observed on KI animals in the fed state is related to the increased food intake and to abnormal lipid profile [46]. In summary, our results show that blocking of S273 phosphorylation in animals fed with a regular (MNT) diet can lead to the development of hyperglycemia and liver damage.

Seeking to investigate the effects of a short-term high-fat high-sucrose (HFHS) diet on WT and KI animals in a simulated obese environment, we evaluated them during and after eight weeks of the diet. This experimental setup was chosen since previous studies showed insulin resistance development in animals treated with HFHS diet [47,48,49]. However, our results did not show any significant changes in ipGTT and ipITT between the two groups. We did observe an increase in white adipose tissue (WAT) weight and a decrease in brown adipose tissue (BAT) weight in the animals on the HFHS diet. A previous study found that PPARγ ligands can promote BAT activation similar to cold exposure by increasing the expression of thermogenic markers (Ucp1 and Cidea) and UCP-1 positive cells in this tissue, which leads to a reduction in BAT mass [12,50]. Increased WAT weight is related to PPARγ activation, which was previously reported to be linked to S273 phosphorylation blockage, promoting coactivators recruitment, and consequent activation of adipogenesis [17].

As well as being observed in MNT diet, our results suggest that KI animals fed with HFHS diet also developed liver injury, since changes on its weight and in glycogen levels were observed. High-fat diets have been associated with reduced liver weight, which in turn leads to decreased hepatic triglycerides and cholesterol levels and increased steatosis, ultimately promoting liver damage [51,52]. Conversely, fasting reduces hepatic glycogen levels through glycogenolysis, while feeding increases glycogen synthesis and hepatic content, regulated by the pancreatic hormones, insulin, and glucagon [53,54]. While the KI group exhibited a regular behavior in the fasted state, their glycogen content during the fed state suggested liver damage and unbalanced pancreatic hormone secretion.

Still, in liver analysis of HFHS fed animals, gene expression analysis showed decreased *PPARγ* levels, consistent with recent reports showing PPARγ as target of epigenetic modifications after high-fat diets, which is demethylated and overexpressed after 12 weeks of HFD [55], and which is hypermethylated and downregulated after longer HFD feeding [56]. The HFHS diet has also been shown to promote epigenetic modifications, leading to changes in the expression of a set of genes involved in metabolism and homeostasis [57,58]. It is important to highlight that in all diets used in our study, KI animals presented decreased *PPARγ* expression, which may be intrinsic of this lineage, but also related to epigenetic modifications.

Aiming at achieving the development of obesity scenario, we tested a prolonged and chronic feeding routine, in which WT and KI animals were fed with high-fat diet (HFD) for 18 weeks. As expected, in this approach the animals presented significant differences in insulin sensitivity, and in weight gain. Interestingly, KI animals gained less weight during the HFD, but they had a higher body weight after weaning. This inverse relationship suggests that higher growth is related to a sensitive sub-population, which is vulnerable to the metabolic disrupting effects [59]. Thus, our KI animals may be more responsive to metabolic changes.

Our study showed that HFD led to the repression of *PPARγ* gene expression in both WT and KI animals, possibly due to obesity-induced DNA hypermethylation [56]. This finding is consistent with previous reports of diabetes mouse models (Lep-/Lep-) that were HDF-treated, in which the repression of PPARγ gene expression was observed in visceral adipose tissues (VAT) [60]. Furthermore, we found that high glucose levels induced mir27-a expression, which downregulated *PPARγ* gene expression [61]. Contrasting with the lower mRNA levels, our results show that PPARγ S273A in the HFD-treated KI animals group presented increased protein content, suggesting that phosphorylation at S273 may regulate PPARγ levels by post-transcriptional mechanisms, as ubiquitination [62,63,64]. In fact, it is known that PPARγ is ubiquitinated by MKRN1 and Siah2 E3 ubiquitin ligases, which target it to proteasomal degradation in adipocytes [65,66]. In this sense, our data indicates that PPARγ phosphorylation interferes in its ubiquitination mechanisms under HFD conditions, reducing it, but this hypothesis should be better investigated.

Another interesting finding in our study is the increased amount of PPARγ in KI animals. Phosphorylation has been shown to affect PPARγ translocation to the cytoplasm, resulting in protein degradation [67]. Despite the contrasting results in terms of PPAR gene expression and protein content in our animals, these findings agree with reports that demonstrate the lack of correlation between gene expression and protein content. Several events, such as transcription, post-transcriptional regulation, translation, post-translation, and protein degradation, can influence total protein content [68,69].

In terms of KI functionality in obese animals, the blockage of Ser 273 phosphorylation promoted decrease in *NR1d2*, *Txnip*, and *Ppm1a* gene expression. This result indicates phosphorylation modulation responses, in which its reduction downregulates dephosphorylation [70]. Additionally, we observed different results regarding Gdf-3 modulation under our experimental conditions compared to previous reports. Gdf-3 has been identified as an important pathway for promoting insulin sensitivity in the S273A *knockin* lineage developed by homologous recombination in mESCs [18], which indicates a promising target for diabetes and obesity modulation. However, our study found that Gdf-3 modulation was different from that previously reported. This difference may be explained by the involvement of PPARγ in promoting Gdf-3 transcription [17,71,72] and its influence on lipolysis and inflammatory response [72,73]. Another investigated target also downregulated in KI mice was *PDK4* (Figure 5E), whose reduced expression may be related to decreased PPARγ expression (Figure 4E), or its involvement with activation of gluconeogenesis [74,75].

Despite the expected positive effects observed in the PPARγ S273 phosphorylation blockage, such as an increase in insulin sensitivity, adiponectin levels, and a decrease in plasma insulin in the fed state, our KI animals presented some negative effects, such as increased eWAT, liver steatosis, and decreased hepatic triglyceride storage. The increased eWAT may be related to the higher activation of PPARγ induced by HFD, which promotes adipogenesis and leads to the expression of PPARγ-target genes related to liver lipid accumulation [17]. Regarding triglyceride content, our results are consistent with some reports which showed this reduction and related it to increased adiponectin levels, decreased *Cd36* and, consequently, causing improvement in insulin sensitivity [76], as observed in our KI animals (Figure 1A–D).

Overall, our observations show that the KI lineage exhibits hyperglycemia, hypoinsulinemia, and higher body weight at weaning, with the potential for liver injury in older age, even when consuming a regular diet. The liver glucose metabolism in KI lineage is altered, with increased *TNFα*, *Cd36*, and *Fabp4*. When fed with HFHS diet, both WT and KI animals did not develop obesity or insulin resistance, but inhibiting PPARγ S273 phosphorylation induced changes in liver metabolism, mainly related to glycogen metabolism and increased WAT. Although prolonged high-fat diet feeding resulted in obesity, our KI lineage is protected from developing insulin resistance, with increased KITT and adiponectin and decreased insulin levels. However, adverse effects were found in this lineage, including increased hepatic steatosis and adipocyte hypertrophy, highlighting that in all conditions decreased *PPARγ* expression was observed. Although blocking S273 phosphorylation could be a promising therapeutic strategy to improve insulin sensitivity, this modification also affects other pathways in the organism, especially those related to steatosis and glycogen metabolism, resulting in unbalanced pancreatic hormones and liver damage. Despite the negative effects of PPAR S273 phosphorylation on insulin sensitization, our results strongly indicate that it also provides positive effects on liver protection and adipogenesis. In an obesogenic scenario induced by HFD feeding, the protection of S273 phosphorylation improved insulin sensitivity. Therefore, our results suggest that selective modulation of PPARγ S273 phosphorylation by ligands should be achieved in the development of new therapies for type 2 diabetes treatment, while considering the potential effects on the liver.

## Figures and Tables

**Figure 1 biomolecules-13-00632-f001:**
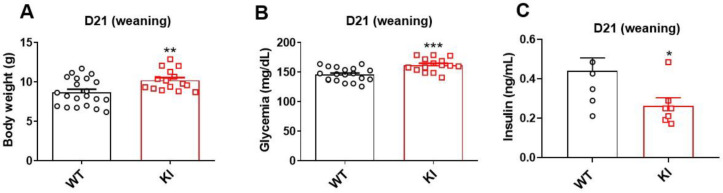
Characterization of KI S273A lineage at weaning (21 days old, D21), comparing with WT animals. (**A**) Body weight (g), (**B**) glycemia, and (**C**) plasma insulin levels in fed state. Data are represented as mean ± SEM. Statistical analysis was performed using non-parametric *t*-test. * *p* < 0.05, ** *p* < 0.01, and *** *p* < 0.001. *n* ≥ 5 per group.

**Figure 2 biomolecules-13-00632-f002:**
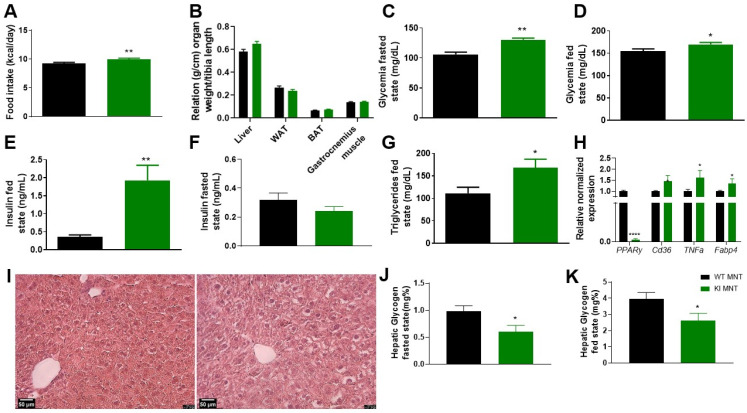
KI S273A lineage after 8 weeks of maintenance diet (MNT), since weaning, comparing with WT animals. (**A**) Food intake during the 8 weeks of MNT diet. (**B**) Relation of organ weight/tibia length on euthanasia day. Glycemia on (**C**) fasted and (**D**) fed states, on euthanasia day. Plasma insulin levels on (**E**) fed and (**F**) fasted states on euthanasia day. (**G**) Plasma triglycerides in fed state. (**H**) Relative normalized expression of *PPARγ*, *Cd36*, *TNFα*, and *Fabp4* on liver, data were normalized to the expression of reference genes: *36b4* and *Rpl27*. (**I**) Liver representative image. (**J**) Hepatic glycogen in fasted and (**K**) fed states. Data are represented as mean ± SEM. Statistical analysis was performed using non-parametric *t*-test. * *p* < 0.05, ** *p* < 0.01, **** *p* < 0.0001. *n* ≥ 5 per group.

**Figure 3 biomolecules-13-00632-f003:**
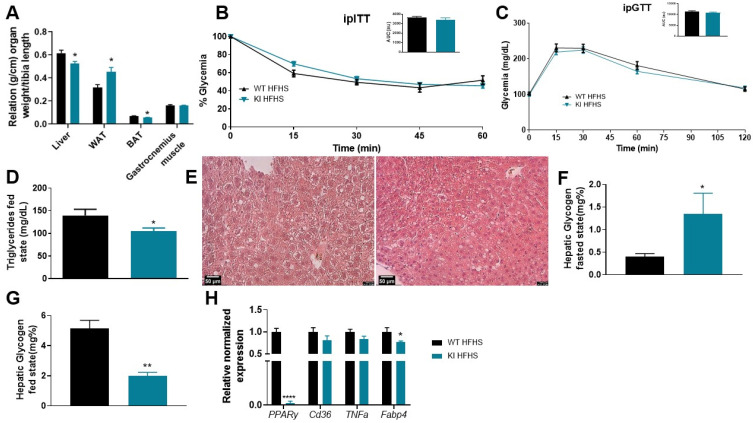
KI S273A lineage after 8 weeks of high-fat high-sucrose diet (HFHS), since weaning, comparing with WT animals. (**A**) Relation of organ weight/tibia length on euthanasia day. (**B**) Intraperitoneal insulin tolerance test (ipITT) with its respective area under the curve (AUC). (**C**) Intraperitoneal glucose tolerance test (ipGTT) with its respective AUC. (**D**) Plasma triglycerides levels in fed state. (**E**) Liver representative image. (F) Hepatic glycogen in fasted and (**G**) fed states. (**H**) Relative normalized expression of *PPARγ*, *Cd36*, *TNFα*, and *Fabp4* in liver, data were normalized to the expression of reference genes: *36b4* and *Rpl27*. Data are represented as mean ± SEM. Statistical analysis was conducted using non-parametric *t*-test. * *p* < 0.05, ** *p* < 0.01, **** *p* < 0.0001. *n* ≥ 5 per group.

**Figure 4 biomolecules-13-00632-f004:**
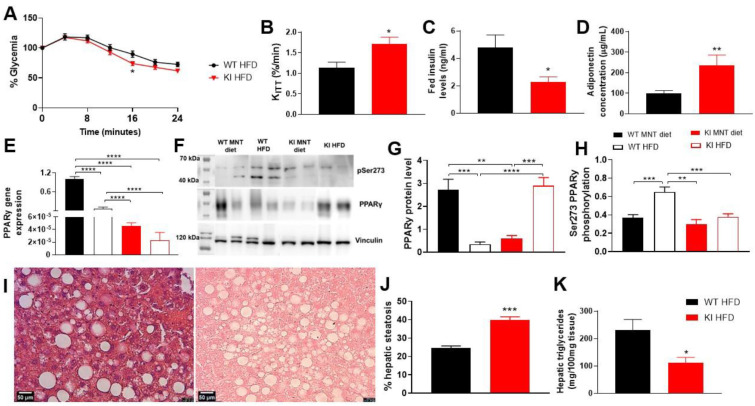
Characterization of KI S273A lineage after 18 weeks of high-fat diet (HFD). (**A**) Percentage of glycemia on insulin tolerance test (ITT) comparing WT and KI at the end of 18 weeks of HFD feeding. (**B**) Glucose disappearance rate (KITT) after ITT, (**C**) plasma insulin levels, and (**D**) plasma adiponectin concentration, comparing WT and KI PPARγ S273A animals. (**E**) Relative normalized expression of *PPARγ* in white adipose tissue (WAT) comparing WT (C57) and KI with MNT and HF diets (HFD). (**F**) Western blotting in eWAT for vinculin, PPARγ, and S273 phosphorylated. Densitometry of bands to quantify (**G**) PPARγ normalized to vinculin and (**H**) S273 phosphorylated, normalized to vinculin. (**I**) Representative liver histological images using hematoxylin-eosin stain of 5 µm paraffin slices, (**J**) quantification of hepatic steatosis using Weka segmentation on Fiji/ImageJ and (**K**) hepatic triglycerides. Data are represented as mean ± SEM. Statistical analysis was performed using non-parametric *t*-test to compare two groups and Dunnett’s T3 multiple comparison ANOVA test. * *p* < 0.05, ** *p* < 0.01, *** *p* < 0.001 and **** *p* < 0.0001. *n* ≥ 3 per group.

**Figure 5 biomolecules-13-00632-f005:**
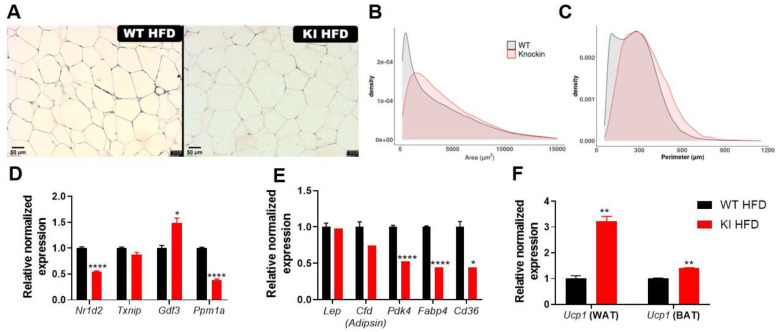
White adipose tissue (WAT) analysis after 18 weeks of high-fat diet feeding. (**A**) WAT histology after embedding it in paraffin and slides were prepared with 5 µm cuts. Analysis of WAT considering density of adipocytes on each (**B**) area and (**C**) perimeter, divided into each group. Relative normalized expression of (**D**) genes regulated by S273 phosphorylation in WAT: *Nr1d2* and *Txnip* described by Choi in 2010, *Gdf3* and *Ppm1a* described in 2020. (**E**) Genes involved on PPARγ and lipid metabolism pathways in WAT: leptin (*Lep*), adipsin (*Cfd*), *Pdk4* (pyruvate kinase dehydrogenase), *Fabp4* (fatty acid binding protein 4), and *Cd36* (scavenger receptor). (**F**) Ucp1 expression in WAT and BAT. Data were normalized to the expression of reference genes: *36b4* and *Rpl27*. Data are represented as mean ± SEM. Statistical analysis was performed using Dunnett’s T3 multiple comparison ANOVA test. * *p* < 0.05, ** *p* < 0.01 and **** *p* < 0.0001. *n* ≥ 5 per group.

## Data Availability

The raw data supporting the conclusions of this article will be made available by the authors, without undue reservation.

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
