# Peer review of "Obesity-Linked PPARγ Ser273 Phosphorylation Promotes Beneficial Effects on the Liver, despite Reduced Insulin Sensitivity in Mice"

_biomolecules, 2023, doi:10.3390/biom13040632_

Round 1
Reviewer 1 Report
In this manuscript, Terra et al reported the generation of PPARy Ser273A mutant mice and characterized its metabolic profiles during weaning and adult stage, with or without HFD for 8 and 16 weeks. The findings are novel and interesting. However, it remains at the level of phenotypic characterization, and lacks mechanistic explanation. There are also some inconsistencies.
Specific Comments
1. The mRNA expression of PPARy seems to be totally gone in KI mice (Fig. 2H, Fig 3H), does this mean that the KI actually resulted in PPARy KO?
2. The discrepancy in insulinemia during weaning and 8 weeks later is intriguing. Hyperglycemia and hypoinsulinemia at weaning suggest that the mice might have difficulty to clear glucose, which could be examined with a glucose tolerance test.
3. Food intake is higher in KI mice, clearly suggesting central effect of this wholebody KI, which was barely discussed.
Author Response
Thanks for reviewer’s 1 comments and for the opportunity to improve our manuscript. Besides we tried to find some mechanistic explanation of the PPARg Ser 273 phosphorylation, our data fit more with phenotypic characterization. The most interestingly findings of our data are the liver responses to S273 phosphorylation, but we could not perform more molecular studies to find the mechanisms of how it happens. It is important to mention that this study set up a new mice lineage and perform a deep characterization of it, bringing a huge dataset of this, I believe that the mechanistic studies are of huge importance to, but it takes another page in this study, and to perform it, we need more resources and time.
Concerning the specific comments:
- The mRNA expression of PPARg is decreased in KI mice, but not totally suppressed. We modify the figures 2H and 3H to better show the graph axis, which shows that PPARg is downregulated in liver. However, it is possible to observe in figure 4E, and 4F that PPARg in KI, besides the downregulation, is present in adipose tissue as mRNA and as protein. Moreover, we also show that the PPAR protein content is higher in KI animals feed with HFD in our western blotting.
- We agree that the discrepancy in insulinemia is intriguing, and we did not imagine finding these results when we performed the first assays with the mice at weaning. I also agree that the glucose tolerance test could clarify this question, however, we did not ask for approval of this test in etic committee in the beginning of our study trying to minimize animal suffering, and we do not had permission to perform this test during this study.
- Concerning food intake, we improved our discussion on it, in page 11, first paragraph.
We also performed an extensive language correction of our text, and now we hope Reviewer 1 consider the improvements of our manuscript as more suitable to be published in Biomolecules.
Reviewer 2 Report
The manuscript claims study of PPAR gamma Serine 273 phosphorylation effects on the liver, despite reduced insulin sensitivity in single S273A mutation (knockin lineage- KI) mice. However, the results of sequencing and Real time PCR seems to show that these claimed KI mice have complete knockout of PPAR gamma expression. To my understanding, the KI mice developed in this study does not form a PPAR gamma with S273A mutation. Instead it mimics a PPAR gamma knock out condition. How do the authors justify involvement of altered protein function if no PPAR gamma protein is being synthesized?
The age and the sex of the mice used in the experiments are missing. It is important to have sex matched mice in each experiment. Data points in graphs are missing.
The diet regime is not clearly explained. The representative write up in methods and the supplementary image requires more clarity.
Time points of glycemic measurements as depicted in the methods section lack clarity.
Statistical tests in methods sections differ from that in figure legends.
Mann-Whitney's test, if performed is not a "t- test".
Overall approach and methods do not suffice for justifying the claim made by the authors.
Author Response
We thank reviewer’s 2 for these comments and apologize if we were not clear enough to explain our results, hypothesis and conclusions. We would like to clarify that DNA sequencing was performed in exhaustion for each animal born, in each crossing between animals, in more than 3 generations, showing the insertion of the mutation in PPAR gene. This DNA sequencing was performed from tail genomic DNA.
Concerning RNA expression and protein content in mice, we observed that KI animals presented downregulation of PPARg, but not a knockout. It is known that a total knockout of PPAR in homozygote animals is not viable to health life, due to animals presents lot abnormalities as lipoatrophy and severe metabolic disturbance (https://doi.org/10.1073/pnas.1314863110). Other studies show that total-body deletion of the two Pparg alleles provoked generalized lipoatrophy along with severe type 2 diabetes. structural and functional alterations of the kidney (https://doi.org/10.1371/journal.pone.0171474).In addition, the first attempts to generate whole body PPARγ knockout (KO) mice showed that loss of PPARγ caused impaired terminal differentiation of the trophoblast and placental vascularization resulting in utero lethality of null embryos and showed lack of adipose tissues, which established the essential role of PPARγ in adipogenesis. Full body PPARγ null mice performed with Cre recombinase preserves PPARγ expression in the trophoblast but only 10% reach adulthood. The characteristics of these mice are: lipodystrophy, organomegaly, decreased leptin and adiponectin in plasma, insulin resistance, elevated free fatty acids (FFAs) and hypotension (doi: 10.3390/ijms17081236). In our animals we did not observe any of these abnormalities, as the animals were phenotypically similar to WT. The mRNA expression of PPARg is decreased in our KI mice. We modify the figures 2H and 3H to better show the graph axis, which shows that PPARg is downregulated in liver. However, it is possible to observe in figure 4E, and 4F that PPARg in KI, besides the downregulation, is present in adipose tissue as mRNA and as protein. Moreover, we also show that the PPAR protein content is higher in KI animals feed with HFD in our western blotting.
We apologize for not be clear enough regarding age, sex, diet regime and other characteristics of our experimental setup. In all our experiments we used male mice. The ages varied depending on the experiment in which mice were at weaning (3 weeks-old) or were feed with MNT diet (starting at 3 weeks-old for 8 weeks), or with HFHS diet (starting at 3 weeks-old for 8 weeks), or with HFD (maintenance diet from 3 weeks-old until 6 weeks-old, FHD started at 6 weeks-old for 18 weeks). Animals were maintained at cages in number to 3-5, on a photoperiod of 12:12 light/dark cycle, at 21-24°C, with free access to food and water. These data were better explained in our text and in the figure of the supplementary material and the number of animals used for each graphic are pointed on the figure captions (page 3, paragraph on methods section “Animals” and “Diets and Experimental design”).
Concerning time points glycemic measurements, we thank the reviewer 2 for the opportunity to improve our manuscript. We performed all the suggested modifications, including time points of glycemia (page 4, methods section).
We apologize for the mistake in statistical methods section. We reviewed the text and put the correct names of statistical tests (page 3, methods section on “Statistical analysis” and in all figure captions).
We again thank reviewer’s 2 for all his comments and for the opportunity to improve our manuscript. Also, we apologize if we were not clear enough to explain our results, hypothesis, and conclusions. We performed a detailed revision in our text and conclusions to improve its quality and to clarify our ideas and findings. We hope now, reviewer 2 consider our manuscript.
Reviewer 3 Report
This manuscript deals on the Obesity-linked PPARg Ser273 phosphorylation promotes beneficial effects on the liver, despite reduced insulin sensitivity in mice. The work has been designed with good scientific knowledge and completed with the required experiments. The outcome of this study result was interesting. This manuscript is perhaps publishable in biomolecules, but there are notable recommendations to enhance the quality of the paper.
The author needs thoroughly check the manuscript and correct the mistakes. Some of the errors are listed below.
1. The Material and methods regarding the animals and experimental protocols need to be more detailed and chemicals with the company name
2. How old were the mice at the beginning of the feeding trial? Weigh? How many mice were there per cage? What was the temperature in the animal house? What is a semi-natural light/dark condition?
3. Check the abbreviation throughout the manuscript.
4. Conclusion-I would recommend focussing on the novelty of this study
5. Discussion on possible mechanisms with recent References
6. Fig. 3E. The author needs to correct the alignment of labeling and add scale bar size.
Author Response
This manuscript deals on the Obesity-linked PPARg Ser273 phosphorylation promotes beneficial effects on the liver, despite reduced insulin sensitivity in mice. The work has been designed with good scientific knowledge and completed with the required experiments. The outcome of this study result was interesting. This manuscript is perhaps publishable in biomolecules, but there are notable recommendations to enhance the quality of the paper.
The author needs thoroughly check the manuscript and correct the mistakes. Some of the errors are listed below.
- The Material and methods regarding the animals and experimental protocols need to be more detailed and chemicals with the company name
- How old were the mice at the beginning of the feeding trial? Weigh? How many mice were there per cage? What was the temperature in the animal house? What is a semi-natural light/dark condition?
- Check the abbreviation throughout the manuscript.
- Conclusion-I would recommend focussing on the novelty of this study
- Discussion on possible mechanisms with recent References
- Fig. 3E. The author needs to correct the alignment of labeling and add scale bar size.
We thank reviewer’s 3 for these comments and apologize if we were not clear enough to explain our results, hypothesis, and conclusions. Also, we would like to thank for the opportunity to improve our manuscript.
As suggested, we performed a profound revision of our manuscript in terms of English language, methodology, explanations and conclusions to try correct our mistakes. Concerning the specific errors pointed by reviewer 3:
- We inserted more details in material and methods section, regarding the animals, experimental protocols, chemicals and equipment’s used in our experiments. We apologize for the lack of clarity in the first version and now, we hope that we solve this question.
- We also improve the details about our animals experiment in the methodology section and, also, in Figure 2 of Supplementary material. Specifically, animals at weaning were 21 days-old, and started to feed with MNT diet of HFHS diet for 8 weeks. In the other group, of HFD, animals were feed with MNT diet from 6 weeks-old for 18 weeks. The weight of animals at weaning was around 10g, and at 6 weeks-old, 18g. we kept 3-5 animals per cage, on a photoperiod of 12:12 light/dark cycle, at 21-24°C, with free access to food and water, as described at page 3.
- We check and correct the abbreviations thorough the text.
- Concerning conclusion, we tried to be more direct, focusing in the novelty of the study, as suggested.
- We tried to discuss more details of possible mechanisms in our text, but, as I did not perform more molecular experiments, we tried not be speculative about it, and for this reason, we did not expose many hypothesis about it. I think that to detail the mechanism of our study would be very interesting, but our focus here was a more phenotypic study and the establishment of new mice lineage, what generated a big dataset. We understand that more details should come in more advanced study, but we did not resources or time to perform this kind of experiment now. I hope that our shy attempts of introduce something in mechanism had improved our discussion now.
- We apologize for the mistake in Figure 3E, now it is corrected.
We also performed an extensive language correction of our text, and now we hope Reviewer 3 consider the improvements of our manuscript as more suitable to be published in Biomolecules.
Round 2
Reviewer 2 Report
The cited papers talk about increased PPAR gamma activity with S273A mutation. Did the authors perform any PPAR gamma promoter activity assay? Because as the graphs in Figure 2 and 3 say, PPAR gamma expression is reduced in the KI samples.
The increased PPAR gamma expression in western blot is observed in figure 4F in HFD group only. In Figure 4, the authors have shown that HFD group has less PPAR gamma gene expression compared to MNT diet group in KI mice. However, the authors have also shown increased Protein expression in the same group. In an alternative way, in HFD group, there is decreased mRNA expression and simultaneous increase in protein expression. The authors have explained this as" Contrasting with the lower mRNA levels, the PPARγ S273A in the HFD-treated KI animals group presented increased protein content, suggesting that even under HFD condition, phosphorylation at S273 may regulate PPAR levels by post-transcriptional mechanisms, as ubiquitination (Daniels et al., 2019, Clague and Urbé, 2010, Videira et al., 2021). In fact, it is known that PPARγ is ubiquitinated by MKRN1 and Siah2 E3 ubiquitin ligases, which target it to proteasomal degradation in adipocytes". However, low mRNA expression can not be post-transcriptionally modified to give higher protein expression.
With my understanding, my response to this revision will be rejection of this manuscript for acceptance.
Author Response
We acknowledge and appreciate the concerns raised by Reviewer 2, and have taken steps to address them. We have revised our manuscript again to make it more transparent and understandable. We have also made further improvements to the language to enhance the manuscript's clarity.
With regard to the major points raised by Reviewer 2, we have added supplementary Figure 2 to better explain our experimental setup. Additionally, we have revised our conclusions to ensure that they are clearer.
Regarding the specific comments raised by Reviewer 2, we would like to address each point individually. One of the cited papers in our work is from our group, where we investigated the impact of the PPAR S273A mutation in a cellular context. We performed gene reporter assays in this study to verify the mutation's activity and observed that the PPAR S273A mutation increased the PPAR's activity even in the absence of rosiglitazone. This suggests that there is increased coactivator recruitment or corepressor derepression, as shown in Figure 1 of our previous work. Moreover, we demonstrated that PGC-1 recruitment was enhanced in PPAR S273A, while NCor and SMRT binding was reduced in comparison to wild-type PPAR, as shown in Figure 3 of our previous work.
In the current paper, we investigated the PPAR S273A mutation in a physiological context, where we introduced this mutation into KI animals. In terms of gene expression, we observed in Figure 4E that PPAR expression was reduced in two contexts: under HFD for both wild-type and KI animals, and in the KI animals. However, the reduced expression of PPAR in these conditions did not have the same effect on protein content. We observed that KI animals fed with HFD exhibited increased protein amounts, while KI animals treated with MNT diet presented low levels of PPAR protein. At the same time, we observed that wild-type mice treated with HFD also presented low levels of PPAR protein. These results do not change the outcomes obtained in this paper, but they highlight the protein behavior in different animals treated with two different diets.
We would like to emphasize that the protein activity does not depend solely on its content in the cell, as other vital proteins such as corepressors and coactivators, as well as DNA response elements, play a crucial role in completing the activation process. The correlation between gene expression and protein content is not always consistent in the same system, as various events in transcription, post-transcriptional regulation, translation, and post-translation may influence the total protein content, as reported in the literature (Vogel C, Marcotte EM. Insights into the regulation of protein abundance from proteomic and transcriptomic analyses. Nat Rev Genet. 2012 Mar 13;13(4):227-32. doi: 10.1038/nrg3185. PMID: 22411467; PMCID: PMC3654667; Schwanhäusser, B., Busse, D., Li, N. et al. Global quantification of mammalian gene expression control. Nature 473, 337–342 (2011). https://doi.org/10.1038/nature10098). In our case, we showed that gene expression did not correspond to the protein amount.
Regarding ubiquitination or other post-translational or post-transcriptional modifications, we suggested that these events may be occurring, preventing protein degradation and causing the protein content to be higher.
We hope that Reviewer 2 will appreciate the efforts that we have made to address their concerns and consider our manuscript for publication.
Reviewer 3 Report
Accept in present form.
Author Response
Thank you for your thoughtful review and your recommendation for publication. We appreciate your time and effort in reviewing our manuscript. We have carefully considered your feedback and made several modifications to improve the clarity and language of our manuscript. Additionally, we have provided further details and explanations regarding our experimental setup in Supplementary Figure 2, in order to better convey our findings and conclusions.
We hope that these changes address any concerns you may have had and enhance the quality of our manuscript. Once again, thank you for your valuable feedback and we look forward to the possibility of having our work published.